# Awareness and Knowledge of Developmental Coordination Disorder Among Healthcare Professionals in the Eastern Province of Saudi Arabia: A Cross-Sectional Study

**DOI:** 10.3390/ijerph21121602

**Published:** 2024-11-30

**Authors:** Abdulaziz A. Al-Ahmari, Abdullah A. Alshabaan, Ali A. Almeer, Mohammed N. AlKhater, Mohammed A. Al-Ibrahim, Hassan H. Altuwal, Alaeddin A. Al-Dajani, Saleh A. Alqahtani, Mohammed A. Al-Omari, Abdullah K. Almutairi, Faisal O. AlQurashi

**Affiliations:** 1Department of Paediatrics, King Fahd Hospital of the University, Imam Abdulrahman Bin Faisal University, Dammam 34224, Saudi Arabia; adajani@iau.edu.sa (A.A.A.-D.); momari@iau.edu.sa (M.A.A.-O.); akalmutairi@iau.edu.sa (A.K.A.); faisal.alqurashi@yahoo.com (F.O.A.); 2College of Medicine, Imam Abdulrahman Bin Faisal University, Dammam 34224, Saudi Arabia; alshabaan.abd@gmail.com (A.A.A.); ali22almeer@gmail.com (A.A.A.); mohmdnazar@hotmail.com (M.N.A.); mohammed3195@gmail.com (M.A.A.-I.); hassan59684@gmail.com (H.H.A.)

**Keywords:** developmental coordination disorder, dyspraxia, motor delay, children, health promotion, public health

## Abstract

Developmental coordination disorder (DCD) is a lifelong neurological disorder impairing the coordination and planning of motor and sensory tasks. Its functional manifestation includes difficulties in various aspects of daily living, making early diagnosis and management essential. This cross-sectional, questionnaire-based study targeted healthcare providers in Saudi Arabia’s Eastern Province who work with children under the age of 18. The questionnaire was completed through field visit interviews and electronically via social platforms from October 2023 to March 2024, collecting data on demographics, professional experience, familiarity with related disorders, and awareness of dyspraxia symptoms. Of the participants, 21.2% had previously diagnosed at least one child with dyspraxia, with family physicians comprising 30% of those diagnoses. The overall mean knowledge score was 17%, ranging from 8.3% to 23.1%. The most recognized symptom was motor learning difficulties (22.7%), followed by gross and fine motor skill delays (22.3%). Notably, 65.15% of respondents were unsure about the gender distribution of dyspraxia. The findings align with international studies, showing significant knowledge gaps among healthcare providers in the Eastern Province. These findings also emphasize the need for targeted health promotion programs, promotional activities, and media involvement for enhancing public health outcomes, early identification, and better management of DCD.

## 1. Introduction

Developmental coordination disorder (DCD) is defined as a lifelong neurological disorder impairing motor and sensory tasks’ coordination and planning [1]. The disorder has been known since the early twentieth century, and Collier James Stanfield first described it as “congenital maladroitness” [2]. Multiple labels have been assigned to the disorder, including dyspraxia, disorder of sensory integration, and clumsy child syndrome [2]. The disorder has an incidence of 6–10% globally, with a male-to-female ratio of 4:1 [2,3,4]. Common risk factors are premature birth and extremely low birth weight [1,5].

Individuals with dyspraxia commonly exhibit a combination of ideational or planning dyspraxia, affecting coordination and planning, and ideomotor or executive dyspraxia, affecting motor activities’ fluency and speed. DCD’s functional manifestation includes difficulties in various daily living aspects. In preschool children, parents often report delays in developmental milestones like crawling, walking, and speaking, along with difficulties in dressing, ball skills, and social interactions. Approximately 25% of children with DCD are referred before starting school, while the remaining 75% are referred in the early primary school years [6]. During this time, issues from preschool persist, such as slow or immature handwriting and difficulty copying from the blackboard. As a result, significant delays often occur before these children receive specialist support. Some may have shown delays in gross motor and language skills, while others may have been perceived as irritable or challenging. When it comes to DCD’s effect, quality of life is a main concern. Children diagnosed with DCD may encounter heightened anxiety, potentially leading to disorders such as social phobia and obsessive-compulsive disorder [7]. Notably, the emotional challenges children with DCD experience can directly contribute to mood issues during adolescence and adulthood, further adversely affecting their quality of life [8].

Blank et al. published international practice recommendations on DCD’s definition, diagnosis, and management in 2012 and updated them in 2019 [2,9]. A DCD diagnosis should be reserved for children under five years of age with severe difficulties due to large differences in typical motor development [2]. Among the parents, teachers, and medical professionals surveyed by Wilson et al., only 20% were familiar with DCD, highlighting the need to increase awareness of the condition [10]. As a result of delayed developmental milestones (e.g., crawling, walking, and speech), Gibbs et al. estimated that 25% of children with this condition are identified before they start school. Many parents and teachers may not be aware of the condition. Children with DCD may be mislabeled or misdiagnosed as slow learners or dyslexic or as having attention-deficit disorder because of their symptoms’ similarity to those of other conditions. These children may also be perceived as not trying hard enough when they are actually putting in a great deal of effort [1]. The main management for DCD is carried out by occupational therapists who provide support for individuals with DCD by assessing and managing children of all ages to improve their social, emotional, behavioral, and motor skills [11]. To our knowledge, this study represents the first audit on the awareness toward DCD among healthcare providers in the Kingdom of Saudi Arabia.

In this study, the aim is to identify gaps in knowledge, measure awareness, and highlight the importance of health promotion and education on DCD among healthcare professionals in Saudi Arabia, specifically in the Eastern Province. Furthermore, the relevant demographic and professional factors that might influence healthcare professionals’ familiarity with the condition are assessed. Adequate basic knowledge about DCD’s signs and symptoms is pivotal in early diagnosis and management. DCD, when not diagnosed early, might lead to delayed milestones and difficulties in learning motor skills. Appropriate programs to educate the relevant practitioners could be implemented, preventing worse long-term outcomes.

## 2. Materials and Methods

This is a cross-sectional questionnaire-based study that was conducted from October 2023 to March 2024. Participants were healthcare providers of Saudi Arabia’s Eastern Province who dealt with children up to the age of 18. They indicated their willingness to participate by providing their informed consent. We excluded developmental-behavioral pediatricians and pediatric neurologists because the condition is part of their assessment scope. The initial calculated sample size was 384, as estimated by the Open-Source Epidemiologic Statistics Calculator [12], with a confidence interval of 95% and a 5% margin of error. The study received 339 responses and resulted in 264 subjects after excluding incomplete questionnaires.

The self-administered questionnaire in this study was created based on previously validated tools for assessing knowledge about DCD [10,13,14]. This revised version of the questionnaire has not been tested for reliability; however, face validity has been covered through the opinion of two experts from the relevant field. Applied amendments to the questionnaire were mainly concerning demographic data (marital status and city of residence). The questionnaire was written in the English language to suit all healthcare providers. Participants were recruited randomly through pre-notification randomized field visits to healthcare centers and hospitals and via electronic social platforms.

Twenty medical students collected data from healthcare providers in different healthcare centers in the province by conducting individual face-to-face interviews or sharing the link for the questionnaire online. The questionnaire was composed of three parts: (a) demographic data (six questions), (b) professional experience (eight questions), and (c) knowledge about DCD (six questions). The mean time for completing the questionnaire was 6.7 min. The demographic data section consisted of six items that were further divided into subcategories for better recall and analysis. Table 1 represents the questionnaire used in this study.

Data were analyzed using Statistical Package for Social Sciences version 21. Categorical variables were presented as percentages, and continuous data were presented as means and standard deviations. All negative statements were reversed before analysis. Simple descriptive analysis was used to measure the study’s variables. Bar charts and tables were used to summarize the study’s figures and results. Comparisons were made among the characteristics of participants. Variables included participant age, gender, marital status, number of children, nationality, place of residence, current practiced profession, location of practice, years of experience, treated age groups in practice, working full or part time, highest held degree, number of children seen per week, number of patients with motor developmental delays seen for the past six months, familiarity with different developmental disorders, including DCD, involvement in treating any of these disorders, previous assignment of DCD as a diagnosis, symptoms of DCD, and proposed actions related to awareness of DCD. Symptoms of DCD were defined based on the Diagnostic and Statistical Manual of Mental Disorders—5th edition (DSM-5).

## 3. Results

This study evaluated healthcare providers’ level of knowledge toward DCD and described its relation to different variables.

### 3.1. Participants Characteristics/Sociodemographic

A total of 264 subjects were included in the study: 57.1% were females, 55.3% were between 20 and 30 years of age, 48.8% were single, 60.6% had no children, and 92.8% were Saudi healthcare providers. Of the total subjects, 32.9% were general practitioners, and 47.7% had been practicing health care for one to five years. A total of 21.2% of providers had diagnosed at least one child with dyspraxia before; the majority of them were family physicians (30%) (Table 2).

### 3.2. Participants Familiarity with Neurodevelopmental Disorders

We decided to compile all medical-related professionals into one group, “medical professionals” (*n* = 240/264), which included general practitioners, pediatricians, family physicians, nurses, surgical subspecialities, and others. The second group was “allied health professionals” (*n* = 24/264), which included occupational therapists and physical therapists. Regarding participants’ background knowledge about common neurodevelopmental and childhood conditions between the two groups, the most known condition among healthcare providers was speech and language disorders (47.3%), followed by medical professionals’ familiarity (48%) and allied health professionals (41.6%). We also tested participants’ knowledge of DCD using its old names to consolidate and/or ensure familiarity, such as clumsy child syndrome, dyspraxia, and motor learning disability. The most recognized terminology for the condition by all participants was DCD (34.4%), followed by motor learning disability (34.4%), dyspraxia (30.6%), and clumsy child syndrome (22.3%). Medical professionals showed a relatively higher familiarity with common neurodevelopmental and childhood conditions as compared to allied health professionals (Table 3).

### 3.3. Participants Familiarity with DCD Features

The mean knowledge score ranged between 8.3% and 23.1%; the overall score was 17%. The most recognized symptom of dyspraxia by study subjects was poor academic performance (23.1%), followed by motor learning difficulties (22.7%) and gross and/or fine motor skills delay (22.3%). Allied health professionals showed more familiarity with DCD symptoms compared to medical professionals, except in the case of poor academic performance (23.8% vs. 16.7%) and suicide risk (18.75% vs. 16.7%) (Figure 1, Figure 2 and Figure 3). The majority of the subjects were unsure about the gender distribution of dyspraxia (65.15%), with approximately 33% of allied health professionals answering correctly compared to 24% of medical professionals.

### 3.4. Participants Views on DCD Management Strategies

Regarding participants’ views of needed strategies for managing DCD, 45.45% of all healthcare providers stated that further research is needed on DCD. The majority agreed on the need for more education around the condition (42.4%). Among the participants, 18.9% stated that it is relatively easy to identify DCD. A total of 22.7% of healthcare providers stated that there are adequate support professionals for children with DCD in the school system (Table 4).

## 4. Discussion

This study assessed the level of knowledge about DCD among healthcare providers in the Eastern Province of Saudi Arabia. All data on the condition were based on previously published international reports. Our study targeted healthcare providers and did not include primary caregivers or teachers.

### 4.1. Comparison with International Findings

The findings revealed an overall knowledge deficit toward DCD among healthcare providers dealing with children in Saudi Arabia’s Eastern Province. One-third of our cohort of healthcare providers were familiar with DCD compared to almost 50% in other international studies [13,14]. Notably, 62.5% of the allied health professionals of our cohort identified gross motor and/or fine motor skills delay as the most common motor feature, while 85% of allied health professionals in Australia identified motor learning difficulties as the most common motor feature [13]. Depression was the least identified feature of DCD, which is in keeping with other international studies [13,14]. This could be due to the lack of adequate education about the condition during academic and professional learning. Our cohort was more familiar with other neurodevelopmental and childhood conditions compared to DCD and its related old terminologies. This observation was in keeping with the findings of Hunt et al. Australian study [13]. Different from other international reports, our cohort was more familiar with the new term “developmental coordination disorder” and the outdated term “motor learning disability” compared to “dyspraxia” [10,13]. Using the term “developmental coordination disorder” consistently is vital, and achieving uniformity in terminology among healthcare providers is the crucial first step toward promptly and accurately identifying DCD.

### 4.2. Medical Professionals vs. Allied Health Professionals

Our findings revealed an overall mean knowledge score of 17% for DCD symptoms among all healthcare providers. Less than one-third of medical professionals correctly identified common motor features of DCD, whereas half of the allied health professionals were successful in identifying these symptoms. This knowledge discrepancy between the two groups is in line with other international reports, with the exception of the poor overall knowledge trend in our cohort compared to that in other reports [13]. DCD is a motor skill acquisition deficit disorder according to the DSM-5 [1], and lacking basic knowledge about its motor symptoms can be problematic in early detection and management. The observation that allied health professionals are more knowledgeable about the motor features of DCD is of no clinical importance in regard to early identification because they occupy the last step along the management ladder. Table 5 compares physicians’ knowledge score differences between our cohort and cohorts from Canada, UK, USA, and Australia.

### 4.3. Knowledge of DCD’s Symptoms

Furthermore, our cohort showed little knowledge of DCD’s nonmotor features compared to its motor features. These include DCD’s psychosocial and emotional outcomes, like anxiety, depression, difficulty building relationships, and poor social skills [7,15]. This finding is similar to that of other international studies [10,13]. Early identification of DCD’s nonmotor features is crucial because they represent social and emotional challenges and may eventually affect the child’s quality of life [16,17]. The absence of both motor and nonmotor skills may result in social stigma, causing the child to feel different from their peers. Also, emotional and psychosocial effects on these children might warrant intervention, which should be considered in their management. The findings of this study sound that almost two-thirds of participants are not familiar with the genders, even though the reported male-to-female ratio is clinically significant at 4:1 [2,3,4]. This lack of awareness could contribute to overdiagnosis on the female side and underdiagnosis on the male side, resulting in ineffective management.

### 4.4. Knowledge Deficits

Our study’s findings showed a significant knowledge deficit among all healthcare providers, which will definitely affect the processes of proper surveillance, diagnosis, and management of cases of DCD. Despite the existence of clear diagnostic criteria and practice recommendations [1,2], augmenting familiarity toward DCD may help in identifying high-risk populations but may not necessarily ensure proper diagnosis. Furthermore, less than one-third of participants thought the DSM-5 contained enough information about DCD for an accurate diagnosis to be made. This may reflect the questionable familiarity of the cohort toward the DSM-5′s contents, lack of exposure to the DSM-5 in their educational programs, inadequate training on the condition, or even general unfamiliarity with the diagnostic approaches. A total of 21% of our cohort believed education is needed for a better understanding of the condition. Family physicians diagnosed most of the cases of DCD in our study, representing a relative relief in the initial identification process and means of surveillance. This could be due to their professional involvement as primary physicians in all primary healthcare centers in Saudi Arabia. This observation is different from those in other studies where allied health professionals identified the majority of DCD cases [10,13]. Despite their critical role in early detection, family physicians may not always have the specialized training to treat DCD effectively. Studies have shown that awareness and knowledge about DCD among general practitioners, including family physicians, can be insufficient, potentially limiting their ability in diagnosing and treating DCD patients. Given overlapping symptoms, family physicians often have difficulty distinguishing DCD from other neurodevelopmental disorders like ADHD. As a result, they may benefit from additional DCD-specific training to enhance diagnostic accuracy and management. For instance, in the United Kingdom, the National Institute for Health and Care Excellence (NICE) has published guidelines that have contributed to increasing physician familiarity with DCD. NICE’s recommendations include training general practitioners (GPs) to identify and refer cases early to specialist care, significantly improving awareness and referral rates [18]. 

Furthermore, in Canada, the “CanChild Centre for Childhood Disability Research” has carried out DCD awareness programs, offering online training modules and resources for primary care physicians. These resources include diagnostic guidelines, case studies, and management strategies to assist physicians dealing with DCD patients in primary care settings. By providing these resources, CanChild has raised awareness and improved early intervention rates, ensuring a more streamlined referral to occupational therapy and educational services [19].

To enhance DCD identification and management in Saudi Arabia, similar approach could be implemented to the local healthcare infrastructure. This study underscores the need for targeted educational programs, which, combined with regional awareness campaigns, could provide family physicians with the knowledge and confidence to manage DCD effectively and improve patient outcomes [10,13].

### 4.5. Recommendations to Raise Awareness

Medical professionals and allied health professionals, like occupational therapists and physical therapists, could work together to improve outcomes for children with DCD. Establishing clear referral pathways and multidisciplinary teams, family physicians can depend on allied health professionals’ expertise for in-depth assessments and targeted therapies. This teamwork would enable family physicians to monitor developmental progress while allied professionals address specific motor and functional challenges. This collaboration could enhance educational initiatives within the community.

It is clear that change measures must be put in place to provide proper education about DCD and to improve early diagnosis and prevent unwanted consequences. International recommendations on the suggested action for translating knowledge into practice include knowledge creation as a first step and then implementation of an action plan as the second step [20]. All healthcare providers in this study play a significant role in the process of detecting DCD, and each has their own degree of involvement. Proposed strategies for improvement include providing parental education about early motor delays and red flags and promoting early seeking of medical advice.

Implementing targeted educational programs to educate healthcare providers who deal with DCD could be a crucial rule on educating current and future healthcare providers, starting from educating medical students in their respective universities by implementing DCD lectures in their curriculum to educating healthcare professionals and allied health professionals by integrating training modules and workshops into the continuing medical education (CME) programs. Moreover, standardized screening protocols in primary healthcare centers could help healthcare providers in recognizing common DCD presentations and early signs. These screening protocols could be organized with a referral pathway to occupational therapists or developmental pediatricians to ensure effective care is provided through a multidisciplinary team approach.

Next, further research on DCD is needed in Saudi Arabia for proper allocation of resources and manpower. Because allied health professionals and family physicians are more knowledgeable about the condition, they can help in the education process of other medical professionals. Augmenting medical and allied health schools’ curriculums with DCD and other common developmental disorders can also help in acquiring early knowledge about these conditions. Last, medical professionals must advocate for the concept of overall child development and specifically toward DCD through their practices and media promotions.

### 4.6. Study Limitations

Although this study represents the first audit on DCD in Saudi Arabia, it carries its own limitations. Firstly, there is the relatively small number of healthcare providers, especially allied health professionals, who participated by completing the study questionnaire. The findings of our study represent a regional consensus rather than a national one, thus preventing generalizability. Secondly, the lack of local studies on the topic drove us to compare our findings with international reports. Thirdly, the questionnaire used for this study was created based on previously validated tools; however, it has not been tested for reliability. Finally, questionnaires are subjected to various degrees of bias based on social acceptance and individual desirability; however, we tried our best to minimize such bias by minimizing personal demographic data extraction.

## 5. Conclusions

Healthcare providers in the Eastern Province of Saudi Arabia have limited overall knowledge of DCD. Significant knowledge gaps have been found across all healthcare providers, including medical and allied health professionals. Further research on DCD is needed for proper allocation of resources, assessing further potential barriers, and evaluating educational and training strategies. Adequate professional and academic education by updating academic curriculums and educational-promoting activities and promotional advocacy by healthcare providers are needed to improve early identification, effective management strategies, and overall quality of life for those affected by DCD.

## Figures and Tables

**Figure 1 ijerph-21-01602-f001:**
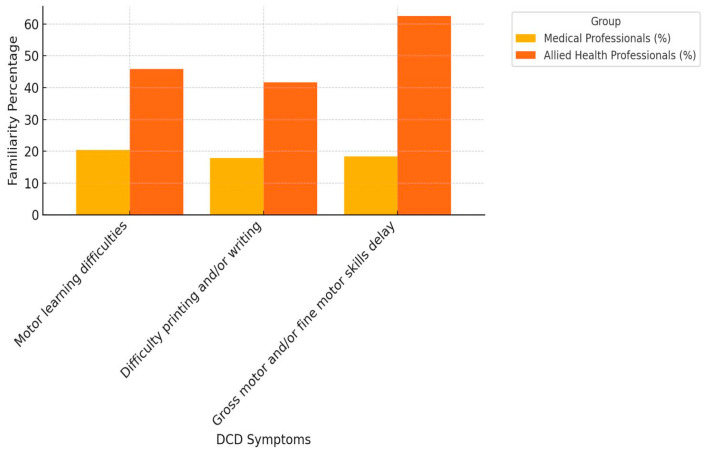
Healthcare providers’ knowledge in identifying common motor DCD features.

**Figure 2 ijerph-21-01602-f002:**
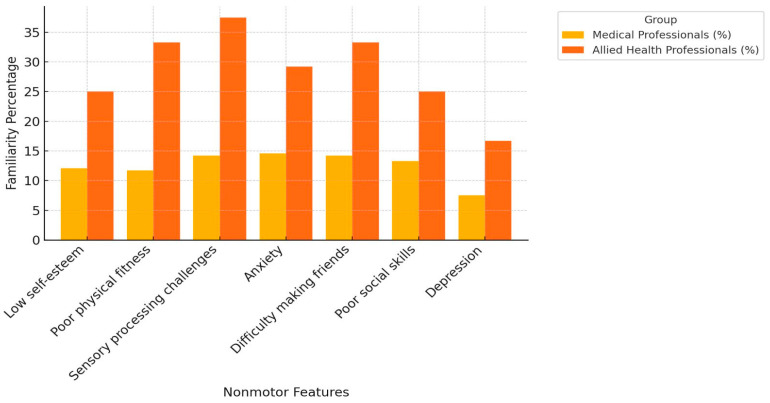
Healthcare providers’ knowledge in identifying common nonmotor DCD features.

**Figure 3 ijerph-21-01602-f003:**
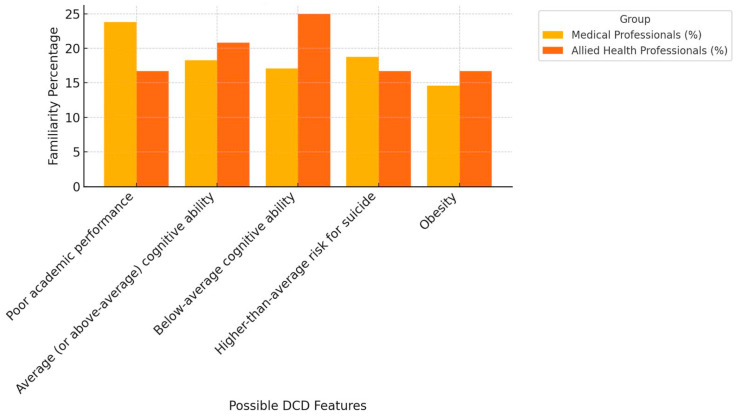
Healthcare providers’ knowledge in identifying “may be” DCD features.

**Table 1 ijerph-21-01602-t001:** The study questionnaire.

Section	Questions	Comment
Demographic data	(1) How old are you? (2) What is your sex? (3) What is your nationality? (4) In which city do you reside? (5) What is your marital status? (6) How many children do you have?	All answer options were in categorical values.
Professional experience	(1) What is your current profession? (2) Where do you currently practice your profession? (3) How many years have you been practicing your profession? (4) Do you work full or part time? (5) Highest degree/qualification held. (6) Which patient group do you treat? Please tick all that apply. (7) In a standard week, approximately how many children (between 0 and 18 years) would be seen in your practice? (8) How many patients with motor developmental delays do you see every six months?	All answer options were in categorical values.
Knowledge about DCD	(1) From here, you will be shown a list of mental and physical health conditions. Please indicate which disorders from the list you are familiar with. In other words, which of these conditions have you heard of (e.g., in training, school, personal life, in the news, etc.): - Global Developmental Delay. - Speech and Language Disorders. - Intellectual Disability. - Autism Spectrum Disorder. - Attention-Deficit/Hyperactivity Disorder. - Developmental Coordination Disorder. - Tourette’s Disorder. - Obsessive-Compulsive Disorder. - Conduct Disorder. - Childhood-Onset Fluency Disorder (Stuttering). - Clumsy Child Syndrome. - Dyspraxia. - Asperger’s Syndrome. - Spina Bifida. - Dyslexia. - Oppositional Defiance Disorder. - Motor Learning Disability. - Chromosomal Disorders. - Learning Disability.	Answers: I have not heard of this condition at all, Very unfamiliar, Somewhat unfamiliar, Somewhat familiar, Very familiar.
	(2) Please indicate which of the following disorders you have been involved in treating: - Global Developmental Delay. - Speech and Language Disorders. - Intellectual Disability. - Autism Spectrum Disorder. - Attention-Deficit/Hyperactivity Disorder. - Developmental Coordination Disorder. - Tourette’s Disorder. - Obsessive-Compulsive Disorder. - Conduct Disorder. - Childhood-Onset Fluency Disorder (Stuttering). - Clumsy Child Syndrome. - Dyspraxia. - Asperger’s Syndrome. - Spina Bifida. - Dyslexia. - Oppositional Defiance Disorder. - Motor Learning Disability. - Chromosomal Disorders. - Learning Disability.	Answers: Yes, No.
	(3) As a medical professional, have you diagnosed a child with Developmental Coordination Disorders (DCD)?	Answers: Yes, No.
	(4) To your knowledge, which of the following do you think are part of the condition of Developmental Coordination Disorder: - Motor learning difficulties. - Difficulty printing and/or writing. - Gross motor and/or fine motor skills delay. - Low self-esteem. - Poor physical fitness. - Sensory processing challenges. - Anxiety. - Difficulty making friends. - Poor social skills. - Depression. - Poor academic performance. - Average (or above-average) cognitive ability. - Below-average cognitive ability. - Higher-than-average risk for suicide. - Obesity.	Answers: Common feature of the condition, May be a feature of the condition, Not part of the condition, Unsure.
	(5) As a healthcare provider, do you agree or disagree with the following statements relating to children and Developmental Coordination Disorder (DCD)? - Further research is needed on DCD. - I feel I need more education/information regarding the condition of DCD. - I believe there are significant benefits from an accurate diagnosis of DCD being given early. - Learning that the estimated incidence of DCD is between 5% and 6% in children would surprise me. - The DSM-5 contains enough information about DCD for an accurate diagnosis to be made. - DCD would be relatively easy to identify. - There are adequate support professionals for children with DCD in the school system. - Accurate diagnoses and classifications are critical for educators to know how to help children with DCD. - I believe educators should play a role in identifying early warning signs that can help to diagnose DCD. - There are too many conditions for educators to keep up with. - Currently, the education system would not be able to adequately support children with DCD due to the lack of knowledge and perceptions about the condition. - I believe there are children in the school system labeled as lazy or defiant that, in fact, have gross and/or fine motor skills impairments. - There are adequate support professionals for children with DCD in the school system.	Answers: Agree, Disagree, Unsure
	(6) What is the typical sex distribution of children with DCD?	Answers: More boys than girls, More girls than boys, Equal between boys and girls, I don’t know.

**Table 2 ijerph-21-01602-t002:** Healthcare providers’ characteristics/sociodemographic (*n* = 264).

Item	Description	Frequency	(%)
Age	20–30 years 31–40 years 41–50 years 51–60 years Above 60 years Total	146 78 34 4 2 264	55.3% 29.5% 12.8% 1.5% 0.75% 100%
Gender	Female Male Total	151 113 264	57.1% 42.8% 100%
Marital status	Single Married Divorced Widow/er Total	129 124 10 1 264	48.8% 46.9% 3.7% 0.4% 100%
Number of children	0 1 2 3 4 More than five Total	160 28 32 20 14 10 264	60.6% 10.6% 12.1% 7.5% 5.3% 3.7% 100%
Nationality	Saudi Non-Saudi Total	245 19 264	92.8% 7.2% 100%
Current profession	General practitioner Pediatrician Family physician Occupational therapist Physical therapist Nurse Surgical subspeciality Radiology Total	87 38 45 9 15 58 7 5 264	32.9% 14.3% 17% 3.4% 5.7% 21.9% 2.6% 1.9% 100%
Years of experience	Less than 1 year 1–5 years 6–10 years 11–15 years 16–20 years More than 20 years Total	43 126 44 29 12 8 264	16.2% 47.7% 16.6% 10.9% 4.5% 3% 100%
Employment type	Full time Part time Total	235 29 264	89% 11% 100%
Highest degree/qualification	Diploma Bachelor Master Specialist/Specialty Certification Consultant/Fellowship PhD Total	6 170 14 41 26 7 264	2.3% 64.3% 5.3% 15.5% 9.8% 2.6% 100%
Age groups covered in practice	Infants (0–1 years) Toddlers (1–3 years) Preschool-aged children (4–5 years) School-aged children (6–14 years) Adolescents (15–18 years) Adults Elderly	96 113 122 128 102 117 73	36.3% 42.8% 46.2% 48.4% 38.6% 44.3% 27.6%
Pediatric patients seen per week	None 1–10 11–20 21–30 31–40 More than 40 Total	64 18 70 51 35 26 264	24.2% 6.8% 26.5% 19.3% 13.2% 9.8% 100%
Patients with developmental motor delays seen in the past six months	None 1–10 11–20 21–30 31–40 More than 40 Total	116 85 32 15 4 12 264	43.9% 32.1% 12.1% 5.7% 1.5% 4.5% 100%
Previous assignment of the DCD diagnosis	Yes No Total	56 208 264	21.2% 78.8% 100%

**Table 3 ijerph-21-01602-t003:** Frequency and percentage of healthcare providers who stated they were “very familiar” or “somewhat familiar” with neurodevelopmental conditions.

Condition *	Total *n* = 264		MP *n* = 240		AHP *n* = 40	
	*n*	%	*n*	%	*n*	%
Global developmental delay	97	36.7%	87	36.2%	10	41.6%
Speech and language disorders	125	74.3%	115	48%	10	41.6%
Intellectual disability	121	45.8%	111	46.2%	10	41.6%
Autism spectrum disorder	127	48.1%	117	49%	10	41.6%
Attention-deficit/hyperactivity disorder	139	52.6%	128	53.3%	11	45.8%
**Developmental coordination disorder**	91	34.4%	80	33.3%	11	45.8%
Tourette’s disorder	104	39.3%	97	40.4%	7	29.1%
Obsessive-compulsive disorder	128	48.4%	121	50.4%	7	29.1%
Conduct disorder	84	31.8%	77	32%	7	29.1%
Childhood-onset fluency disorder (Stuttering)	96	36.3%	85	35.4%	11	45.8%
**Clumsy child syndrome**	59	22.3%	51	21.2%	8	33.3%
**Dyspraxia**	81	30.6%	70	29.1%	11	45.8%
Asperger’s syndrome	74	28%	65	27%	9	37.5%
Spina bifida	119	45%	107	44.5%	12	50%
Dyslexia	95	35.9%	83	34.5%	12	50%
Oppositional defiance disorder	77	29.1%	69	28.7%	8	33.3%
**Motor learning disability**	91	34.4%	80	33.3%	11	45.8%
Chromosomal disorders	123	46.5%	112	46.6%	11	45.8%
Learning disability	125	47.3%	116	48.3%	9	37.5%

* Note. DCD and associated terms for this disorder are in bold emphasis. Abbreviations: MP: medical professionals; AHP: allied health professionals.

**Table 4 ijerph-21-01602-t004:** Participants’ views on needed strategies around DCD management.

Statement	Number of Agree Responses (*n* = 264)	%
Further research is needed on DCD.	120	45.45%
I feel I need more education/information regarding the condition of DCD.	112	42.4%
I believe there are significant benefits from an accurate diagnosis of DCD being given early.	119	45.1%
Learning that the estimated incidence of DCD is between 5% and 6% in children would surprise me.	91	34.5%
The DSM-5 contains enough information about DCD for an accurate diagnosis to be made.	60	22.7%
DCD would be relatively easy to identify.	50	18.9%
There are adequate support professionals for children with DCD in the school system.	52	19.7%
Accurate diagnoses and classifications are critical for educators to know how to help children with DCD.	100	37.9%
I believe educators should play a role in identifying early warning signs that can help to diagnose DCD.	117	44.3%
There are too many conditions for educators to keep up with.	89	33.7%
Currently, the education system would not be able to adequately support children with DCD given the lack of knowledge and perceptions about the condition.	95	36%
I believe there are children in the school system labeled as lazy or defiant that, in fact, have gross and/or fine motor skill impairments.	105	39.8%
There are adequate support professionals for children with DCD in the school system.	60	22.7%

Abbreviations: *n*: number of participants; %: percentage; DCD: developmental coordination disorder; DSM-5: Diagnostic and Statistical Manual of Mental Disorders, 5th edition.

**Table 5 ijerph-21-01602-t005:** Percentages of physicians who correctly identified features of DCD among three different cohort groups.

Item #	Saudi Arabia (Eastern Region)—MP (%)	Saudi Arabia (Eastern Region)—AHP (%)	Australian—MP (%) [13]	Australian—AHP (%) [13]	Canada, UK, US Study—Physicians (%) [10]
**Common motor features of DCD**					
Motor learning difficulties	20.4	45.8	52	85	79
Difficulty printing and/or writing	17.9	41.7	56	75	77
Gross motor and/or fine motor skills delay	18.3	62.5	52	81	70
Mean knowledge score	18.87	50	53.3	80.3	75.3
**Common nonmotor features of DCD**					
Low self-esteem	12.1	25	33	50	59
Poor physical fitness	11.7	33.3	22	38	49
Sensory processing challenges	14.2	37.5	19	22	42
Anxiety	14.6	29.2	22	23	40
Difficulty making friends	14.2	33.3	15	13	40
Poor social skills	13.3	25	13	11	37
Depression	7.5	16.7	7	7	28
Mean knowledge score	12.9	28.5	18.71	27.71	42.14
**May be a feature of DCD**					
Poor academic performance	23.8	16.7	19	55	41
Average cognitive ability	18.3	20.8	11	36	48
Below-average cognitive ability	17.1	25	4	41	16
Higher risk for suicide	18.75	16.7	4	49	20
Obesity	14.6	16.7	4	44	17
Mean knowledge score	18.11	19.18	8.4	45	28.4

Abbreviations: MP: medical professionals; AHP: allied health professionals; DCD: developmental coordination disorder; UK: United Kingdom; US: United States.

## Data Availability

The raw data supporting the conclusions of this article will be made available by the authors on request.

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
