# Peer review of "Awareness and Knowledge of Developmental Coordination Disorder Among Healthcare Professionals in the Eastern Province of Saudi Arabia: A Cross-Sectional Study"

_ijerph, 2024, doi:10.3390/ijerph21121602_

Round 1
Reviewer 1 Report
Comments and Suggestions for Authors
This study examines the awareness and knowledge of Developmental Coordination Disorder (DCD) among healthcare providers in Saudi Arabia's eastern province. Using a cross-sectional survey, it assesses healthcare professionals’ familiarity with DCD symptoms, management strategies, and the challenges they face in recognizing the disorder. Findings indicate significant knowledge gaps, especially among general practitioners, highlighting the need for targeted education and resources to improve early diagnosis and management of DCD in Saudi Arabia. While this topic is important and provides important insights in relation to DCD, the manuscript requires major revision and cannot be accepted in its current form. Below are my recommendations to the authors that could potentially help improve the manuscript.
Introduction and Literature Review: The introduction is well-structured and provides a broad overview of Developmental Coordination Disorder (DCD). However, there is redundancy in presenting historical and international context. Condense some information to focus more on DCD's relevance in Saudi Arabia, linking the local need for awareness and intervention.
Method: I think the method is the weakest part of this article and requires most improvement. The main part of the method which is the questionnaire is not reported and described in its entirety.
1. The questionnaire is the most important part of this study. It must be included in the article. It is important to know what were the exact questions on the survey. The questionnaire should be included in its original form in the article.
2. The process by which the questionnaire was developed must be reported. Once again, the questionnaire is the main aspect of the paper and should be reported in detail.
3. Who developed the questionnaire? How was it validated? Was it pilot tested? What language was the questionnaire in? Were all the participants familiar with the language of the questionnaire.
4. If previously validated tools were adapted, specify the validation process or provide references for each tool used.
5. More detail on random sampling procedures and criteria for including allied health professionals vs. medical professionals is recommended.
6. Was reliability testing conducted? If so, how and what were the results? If not, it is a major limitation since self reported survey often have reliability problems.
Results: Currently, a lot of statistical data in table form is presented which is difficult to interpret. I highly recommend using graphs to interpret this data. Moreover, it may be beneficial to add a comparative table that juxtaposes findings from similar international studies to highlight Saudi Arabia’s unique context.
Discussion: Currently, the discussion is reiterating the results. However, discussion can be made stronger if the authors provide ways to improve awareness for DCD. It could benefit from explicitly connecting the study’s results to actionable recommendations. Include specific interventions, such as targeted training programs or policy changes, to enhance healthcare provider knowledge of DCD.
Author Response
thank you for your review, you can find our response in the uploaded file titled "Cover letter -Reviewers' comments''

Reviewer 2 Report
Comments and Suggestions for Authors
This is a valuable study of a disorder not well-researched nationally in Saudi Arabia.
In lines 74-79: You included all healthcare professionals who dealt with children up to age 18 years in the Eastern Saudi region. How did you end up with only 300 participants as a minimum effective sample size? What was your original population?
In line 80: You said you interviewed 339 participants and ended up with 264 because of incomplete questionnaires. How did you interview participants and have incomplete questionnaire? This only happens if you distribute the electronic questionnaire?! Please clarify.
In line 82: Your questionnaire was CREATED based only previous questionnaires. This means your questionnaire is a new questionnaire (any changes, even for one item, to a validated questionnaire require new validation). Your NEW questionnaire needs to go through through validation through a separate piloting process. You did not report any validation process of the newly developed questionnaire? Why? How to ensure that the participants understood the questions correctly. The incomplete questionnaire that you received could be because the participants faced difficulties understanding the questions!
In lines 83-84: You said the participants were recruited randomly through field visits to health-care centers and hospitals and via electronic social platforms. You have two ways of data collection (interview + electronic questionnaire). This should be clearly reported in the manuscript without only reporting that the participants were interviewed or the distribution of an electronic survey (as reported in the abstract).
In line 201, typo: change (.) to (,) before word augmenting.
In lime 217: Further research was reported, while there was no reporting of the study limitations.
In 224 (conclusion): You said healthcare providers in Saudi Arabia have limited overall knowledge of DCD. How is this correct while your participants were only from one region (the Eastern region)? Also, your sample size is small (considering the two methods of data collection) to generalize your findings to all healthcare providers across Saudi Arabia. Please consider revising all the "overgeneralization" of your data on all healthcare providers in Saudi Arabia throughout the manuscript. It has to be limited to those practicing in the Eastern region only.
Reference: This section could be improved by adding recent national/international publications related to DCD.
Author Response

(The authors gave the same response as above.)

Reviewer 3 Report
Comments and Suggestions for Authors
The article presents a valuable study regarding the awareness and knowledge of developmental coordination disorder (DCD) among healthcare professionals in Saudi Arabia.
Comments
The abstract you've provided presents a valuable study regarding the awareness and knowledge of developmental coordination disorder (DCD) among healthcare professionals in Saudi Arabia. Below are my comments.
Strengths: The study identifies a specific issue—awareness of DCD among healthcare professionals—making it relevant to public health and pediatric care. The use of quantitative measures, such as the percentage of professionals who diagnosed dyspraxia and the overall mean knowledge score, provides a solid basis for conclusions. Conducting the study in 2023–2024 allows for contemporary data, which is essential for addressing current gaps in knowledge and practice.
Meanwhile, some issues as stated below need to be addressed. By addressing these issues, the manuscript can be significantly strengthened, and the study can offer greater insight into the gaps in knowledge and awareness about DCD in Saudi Arabia.
Introduction
The introduction provided offers a thorough background on developmental coordination disorder (DCD) and its implications, especially regarding its prevalence, diagnosis, and impact on children's quality of life. However, a few areas could benefit from improvement to enhance clarity, structure, and depth. Below are my comments:
- The introduction effectively covers key aspects of DCD, including its definition, history, incidence, and major challenges associated with its diagnosis and management. It provides useful global context and historical perspective. The mention of emotional and psychological effects, like anxiety and mood issues, highlights the far-reaching consequences of DCD beyond just motor skills, which broadens the scope of understanding. The paragraph addresses the need for greater awareness, particularly in Saudi Arabia, which connects well to the study's goals.
Please address the following in the introduction:
1. The introduction includes too many details about various aspects of DCD all at once, which can overwhelm readers. For example, the explanation of the symptoms in preschool and primary school-aged children could be streamlined to improve clarity.
2. The statement regarding the delay in receiving specialist advice and the proportion of children referred before or after starting school is repeated twice (once in the early part of the introduction and again later). Consolidating these points will help avoid redundancy.
3. While the study aims to explore gaps in knowledge in Saudi Arabia, particularly in the eastern province, most of the introduction focuses on international contexts and data. The inclusion of more region-specific information early on would ground the reader in the setting of the research more effectively.
4. The brief mention of Saudi Arabia’s occupational therapy centers seems isolated and not well integrated into the larger narrative. This could either be expanded to provide more context or omitted, especially if it doesn't contribute directly to the core message of the introduction.
Materials and Methods
The "Materials and Methods" section you've provided presents a well-structured description of the study design, data collection, and analytical approach. However, there are some areas where clarity, depth, and methodological rigor could be improved.
- Sampling Method:
- While participants were recruited randomly, there is no mention of how randomization was ensured. Were specific randomization techniques (e.g., random number generation, stratified sampling) used during field visits or online recruitment? This would clarify how bias was minimized.
- Questionnaire Validation:
- Validation of the Tool: Although the questionnaire was based on previously validated tools, it is important to describe any modifications made or how the tool was adapted to the local context (Saudi Arabia). Additionally, specifying how the tool was validated for this particular study population would add to the methodological rigor.
- Was the questionnaire pilot tested in the study setting to ensure it was understandable for the target audience? Mentioning this would add credibility to the instrument's applicability.
- Participant Details:
- Participant Distribution: There is no breakdown of how many participants were recruited from each method (field visits vs. online). This information is important for assessing potential recruitment biases. For example, healthcare professionals recruited online may have had different availability or interest compared to those approached in person.
- Exclusion Criteria Beyond Specialists:
- Incomplete Questionnaires: While incomplete questionnaires were excluded, it would be useful to describe what constituted "incomplete" and how many were discarded. This ensures transparency in the data cleaning process.
- Variables and Their Analysis:
- Variable Definition and Analysis: While the study collects a wide range of demographic and professional variables, there is no mention of how these variables were compared (e.g., chi-square tests, logistic regression, t-tests). Were any inferential statistics used to compare knowledge levels between groups? This information would provide a deeper understanding of the analysis.
- Handling Confounders: The section does not mention how potential confounders, such as years of experience or healthcare provider specialty, were controlled in the analysis. This is crucial for ensuring the findings are robust and not due to confounding factors.
- Generalization of the Findings:
- The fact that the study was limited to the eastern province is clear, but how representative this province is of the rest of Saudi Arabia's healthcare professionals is not addressed. This limitation should be acknowledged, and suggestions for future studies that could cover more regions could be provided.
The results
The "Results" section provides a detailed analysis of the study findings, but some aspects of clarity, depth of analysis, and structure could be improved.
- While descriptive statistics are presented, there is no mention of whether the differences observed (e.g., between medical and allied health professionals) are statistically significant. The inclusion of p-values or confidence intervals would help readers understand the strength of these findings. For example, is the difference between medical and allied health professionals’ knowledge of DCD features statistically significant?
- Redundant Statements: Some information is repeated unnecessarily. For example, the statement "We double tested participants’ knowledge about their familiarity with the condition of DCD by using some of its other names..." could be stated more concisely. Consider streamlining these points to avoid redundancy.
- Clarity in Terminology: It’s important to ensure clarity when discussing different terms for DCD, as not all readers will be familiar with the various terminologies (e.g., clumsy child syndrome, motor learning disability). The section could benefit from a clearer explanation of these terms at the outset.
- Gender Distribution Results: The finding that 65.15% of participants were unsure about the gender distribution of dyspraxia is significant but not discussed further. Given that DCD has a reported male-to-female ratio of 4:1, this lack of knowledge is concerning and warrants a deeper discussion of its implications for diagnosis and treatment in practice.
Discussion:
The "Discussion" section presents important insights into the study's findings regarding healthcare providers’ knowledge of developmental coordination disorder (DCD) in Saudi Arabia. However, there are several areas where the argument could be strengthened, clarified, or expanded.
1. Organizing the discussion into clearer subsections (e.g., "Knowledge Deficits," "Comparison with International Findings," "Implications for Practice," and "Recommendations") could improve readability and make it easier for readers to follow the argument.
2. While the discussion notes that only one-third of the cohort was familiar with DCD compared to nearly 50% in other studies, it lacks specific details about those studies. Including specific citations or statistics from these studies would strengthen the argument and provide a clearer benchmark for evaluating the results.
3. The discussion mentions that less than one-third of participants believe the DSM-5 provides enough information for accurate diagnosis. This finding is significant but not explored in depth. It would be beneficial to discuss why participants might feel this way. Is it due to inadequate training, lack of exposure to the DSM-5 in their educational programs, or a general unfamiliarity with the diagnosis process?
4. Expand on the implications of healthcare providers’ unfamiliarity with the psychosocial and emotional aspects of DCD. Why is it particularly concerning that these features are not recognized? Discuss the potential long-term effects on the children and families involved.
5. The discussion states that family physicians diagnosed most DCD cases in this study, suggesting a relative relief in identification. However, it would be beneficial to elaborate on this point. Are family physicians adequately equipped to manage these cases, or do they also require additional training? What challenges do they face in this role?
6. Discussing the potential for collaboration between family physicians and allied health professionals could enhance this section. How might these groups work together to improve outcomes for children with DCD?
7. The recommendations provided are essential but could benefit from further specificity. What types of education programs should be implemented, and who should lead them? For instance, mention the role of universities, professional organizations, or government agencies in facilitating education.
8. Consider expanding on the proposed strategies for improvement by providing examples of successful interventions or programs in other regions or countries that have effectively increased awareness and knowledge of DCD among healthcare providers.
Conclusions
The "Conclusions" section succinctly summarizes the key findings of the study and emphasizes the need for improved knowledge among healthcare providers regarding developmental coordination disorder (DCD). However, there are opportunities to enhance the clarity and impact of the conclusions.
- The conclusion mentions the need for "adequate professional and academic education," but it could be more impactful by specifying what this education should entail. For example, should it focus on specific aspects of DCD, such as diagnosis, management, or the psychosocial impacts? Providing clarity on this could help guide future educational initiatives.
- Expanding on the implications of these findings could enhance the conclusion. For instance, discuss how improved knowledge of DCD could lead to better outcomes for children and families affected by the disorder. This could include improvements in early diagnosis, effective management strategies, and overall quality of life for those impacted by DCD.
- The conclusion could briefly mention the importance of future research in this area, including potential studies that could further explore the barriers to knowledge and the effectiveness of educational interventions. This would help highlight that while the current study has addressed a critical issue, ongoing investigation is necessary for continued improvement.
- Consider mentioning the importance of collaboration between different types of healthcare providers in improving awareness and management of DCD. This could include partnerships between medical professionals and allied health providers to enhance knowledge sharing and comprehensive care for affected children.
Author Response

(The authors gave the same response as above.)

Round 2
Reviewer 1 Report
Comments and Suggestions for Authors
The authors have addressed all my concerns and comments that I provided in my first review. I am happy to see that the the questionnaire used for the data collection is now part of the manuscript. The addition of graphs, rewriting some discussion and adding more context in the introduction do improve the manuscript. However, I still have the same concern related to the questionnaire as before and that is related to its validity and reliability. It seems like the questionnaire was not developed in norms with the typical validity and reliability standards used in the field. This should be addressed in the manuscript- the authors should provide a more transparent picture on how the questionnaire was developed and if its validity and reliability were not tested then it should be reported as a limitation. There should be caution mentioned in the manuscript for future studies on using this questionnaire directly.
Author Response
Comment 1: The authors have addressed all my concerns and comments that I provided in my first review. I am happy to see that the the questionnaire used for the data collection is now part of the manuscript. The addition of graphs, rewriting some discussion and adding more context in the introduction do improve the manuscript. However, I still have the same concern related to the questionnaire as before and that is related to its validity and reliability. It seems like the questionnaire was not developed in norms with the typical validity and reliability standards used in the field. This should be addressed in the manuscript- the authors should provide a more transparent picture on how the questionnaire was developed and if its validity and reliability were not tested then it should be reported as a limitation. There should be caution mentioned in the manuscript for future studies on using this questionnaire directly.
Response: Thank you for your comment. We agree with this point. The following sentence has been added to the methods section (Page 3, Line 101, highlighted): This revised version of the questionnaire has not been tested for reliability; however, face validity has been covered through the opinion of 2 experts from the relevant field.
Also, the following sentence has been added to the study limitation section (Page 17, Line 326, highlighted): Thirdly, The Questionnaire used for this study was created based on previously validated tools, however it has not been tested for reliability.
Reviewer 3 Report
Comments and Suggestions for Authors
Thanks for addressing the delivered comments
Author Response
Thank you for your valuable inputs.